# Radiotherapeutic Strategies to Overcome Resistance of Breast Cancer Brain Metastases by Considering Immunogenic Aspects of Cancer Stem Cells

**DOI:** 10.3390/cancers15010211

**Published:** 2022-12-29

**Authors:** Katharina Hintelmann, Cordula Petersen, Kerstin Borgmann

**Affiliations:** 1Department of Radiotherapy and Radiation Oncology, University Medical Center Hamburg-Eppendorf, 20246 Hamburg, Germany; 2Laboratory of Radiobiology and Experimental Radiooncology, Center of Oncology, University Medical Center Hamburg-Eppendorf, 20246 Hamburg, Germany

**Keywords:** breast cancer, brain metastases, BCBM, stereotactic radiotherapy, WBRT, CTC, radiation resistance, CSC, DNA repair

## Abstract

**Simple Summary:**

Modern radiotherapy offers several options for the treatment of brain metastases from breast cancer. The radioresistant subpopulation of cancer stem cells (CSCs) pose a particular challenge to a complete cure. This is attributable to the enhanced activation of molecular defense mechanisms that prevent cell death as a consequence of DNA damage. Another fundamental feature of CSCs is their evasion of the immune system. Combining inhibitors of both properties with irradiation may be an attractive option to advance existing therapies, and this is the subject of the data summarized here.

**Abstract:**

Breast cancer is the most diagnosed cancer in women, and symptomatic brain metastases (BCBMs) occur in 15–20% of metastatic breast cancer cases. Despite technological advances in radiation therapy (RT), the prognosis of patients is limited. This has been attributed to radioresistant breast cancer stem cells (BCSCs), among other factors. The aim of this review article is to summarize the evidence of cancer-stem-cell-mediated radioresistance in brain metastases of breast cancer from radiobiologic and radiation oncologic perspectives to allow for the better interpretability of preclinical and clinical evidence and to facilitate its translation into new therapeutic strategies. To this end, the etiology of brain metastasis in breast cancer, its radiotherapeutic treatment options, resistance mechanisms in BCSCs, and effects of molecularly targeted therapies in combination with radiotherapy involving immune checkpoint inhibitors are described and classified. This is considered in the context of the central nervous system (CNS) as a particular metastatic niche involving the blood–brain barrier and the CNS immune system. The compilation of this existing knowledge serves to identify possible synergistic effects between systemic molecularly targeted therapies and ionizing radiation (IR) by considering both BCSCs’ relevant resistance mechanisms and effects on normal tissue of the CNS.

## 1. Introduction

Breast cancer accounts for over 30% cancer cases among females and is therefore the most frequently diagnosed malignancy in women [1,2]. A major challenge in the management of this disease is its propensity for brain metastases. In metastatic breast cancer, symptomatic metastases occur in about 15–20% of cases, representing a significant cause of morbidity and mortality [3]. Unfortunately, there is still no cure for metastatic breast cancer, but the evolving treatment options extend life expectancy [4]. Nevertheless, patients with brain metastases still have an unfavorable prognosis [5].

### 1.1. From the Primary Tumor Site to the Brain—The Metastatic Cascade

Metastasis is a complex process that describes the migration of tumor cells and their colonization of a distant organ. It was first summarized by Paget’s famous “seed and soil” hypothesis that describes the interactions between tumor cells and host organ [6]. In the following decades, this process was extensively studied, revealing more details of the metastatic cascade and specific feature of the central nervous system. In 2019, Welsh et al., summarized the hallmarks of metastasis in motility and invasion, as well as the modulation of the microenvironment, plasticity, and colonization [7].

It is believed that only a small subpopulation of cells from the primary tumor has the capacity to form brain metastases. The underlying processes have been described extensively [8]. They are not the central topic of this article and are only briefly summarized here, with particular emphasis on the role of so-called cancer stem cells (CSCs) [9]. One central step from the primary tumor cell to distant metastasis is the epithelial-to-mesenchymal transition (EMT), when cancer cells are converted to migratory and invasive cells. EMT-inducing transcription factors seem to be key components of this dedifferentiation, but there is also evidence of other regulatory mechanisms such as epigenetic modifications, microRNAs, and EMT-associated alternative splicing events. It is largely understood that, in this process, cells can also gain stem cell characteristics or tissue stem cells can progress to cancer stem cells [10]. 

Circulating tumor cells (CTC) can reach the brain via blood vessels, where they must extravasate, and they can be divided by metastatic capacity into CTCs that only gain migratory features and CTCs that are metastasis-competent [11]. Additionally, there are studies demonstrating the existence of a CTC subpopulation, with putative stem cell progenitor phenotypes in patients with metastatic breast cancer [12]. Bryan et al., summarized the known molecular mechanisms in breast cancer cells to survive this journey and successfully colonize the brain, with a particular emphasis on breast-cancer-subtype-specific factors. Many of the alterations linked to breast cancer brain metastases (BCBM) formation summarized in this review are genes/molecular markers related to CSC maintenance and DNA repair, which are essential factors in radioresistance [13].

The CNS represents a unique environment compared to other metastatic sides, and it is characterized by the blood–brain barrier (BBB) and a microenvironment that is shaped by the CNS immune system. The BBB is a selective, semipermeable boundary of endothelial cells, regulating the exchange of ions, molecules, and cells and thus the homeostasis of the CNS [14]. CNS vessels are non-fenestrated and present an obstacle for drugs [15]. Since trans- and paracellular solute movement is very limited, transport is mainly restricted to transporters expressed in CNS endothelial cells—efflux transporters and specific nutrient transporters [16,17]. In developing new systemic treatment options for BCBM, CNS bioavailability is another obstacle that must be overcome for successful treatment.

In addition, the BBB can prevent tumor cell extravasation into the CNS. Four genes have been identified as mediators of tumor cell passage through the BBB, namely cyclooxygenase 2, heparin-binding EGF-like growth factor, alpha 2,6, sialyltransferase [18], and β4 integrin [19]. Furthermore, cancer-derived extracellular vesicles, mediators of cell–cell communication via the delivery of proteins and microRNAs, trigger the breakdown of the BBB. Importantly, micoRNA-181c promotes the destruction of the BBB through the abnormal localization of actin via the downregulation of its target gene, 3-Phosphoinositide Dependent Protein Kinase 1 [20].

The interaction of the breast cancer cell with the microenvironment represents the connection to the establishment of metastases in the CNS niche. Communication with the distant tissue via soluble factors and extracellular vesicles [21] leads to organ-specific changes that occur before metastasis. One of the most important mechanisms in breast cancer metastasis to the brain is the alteration in glucose metabolism [22]. Glucose is the primary energy substrate in the mammalian brain, with astrocytes and neurons being the main consumers. In this interplay, the brain microenvironment influences cancer cells. In the context of CSCs, they are thought to establish their niche by collaborating with astrocytes. Tumor-associated astrocytes have been found to be activated by cyclooxygenase 2 and prostaglandins, followed by the release of chemokine (C-C motif) ligand 7, which in turn promotes the self-renewal of tumor-initiating cells in the brain [23]. During this metastatic progression, immunosuppressive mechanisms are critical in preventing the recognition and destruction of cancer cells by the immune system. In this context, the activation of microglia by breast cancer cells must be suppressed to allow tumor growth. Neurotrophin-3 expressed by breast cancer cells has been found to be a possible mechanism [24]. The complex interactions of the immune system in breast cancer represent one of the most rapidly developing areas of research in this disease, the details of which are discussed below from a clinical perspective.

### 1.2. Brain Metastasis Incidence Depending on Breast Cancer Subtype

In the clinical routine, four biological subtypes are highly relevant: Luminal A, luminal B, human epidermal growth factor receptor-2-enriched (HER2-positive), and triple negative (TNBC). This classification is based on the expression of estrogen and progesterone hormone receptors (ER and PR), HER-2, and histochemical marker Ki67 [25,26]. Receptor expression is a relevant prognostic factor, not only at initial diagnosis, but also for predicting the risk of developing brain metastases [27]. In a large meta-analysis, Kuksis et al., reported a brain metastases incidence in metastatic breast cancer of 31% for the HER2+ subgroup and 32% for the TNBC subgroup, compared to 15% among patients with the HR+/HER2− subtype; these findings highlight the high incidence of brain metastases among patients with HER2+ and TNBC breast cancer and suggest a screening program for these populations [28]. In addition to this, young age, primary tumor size, and nodal involvement are also associated with an increased risk of developing brain metastases [29].

The prognosis of brain metastases can be accessed by the Graded Prognostic Assessment, which was first published in 2008 [30]. According to the latest update in 2020, extracranial metastases and the number of brain metastases were found to be significant in conjunction with established factors, such as patient age, Karnofsky Performance Status, and molecular subtype. In this study, survival ranged between 6 and 36 months [31]. Concerning the molecular subtype, there was clear evidence of a worse outcome in patients with HER-positive and triple negative brain metastases [32]. Comparing different sites of breast cancer metastasis, the overall survival for stage IV breast cancer was lowest regarding brain metastasis, while it was best for bone metastasis [33,34].

The subtype of distant metastases can differ compared to the primary tumor. A recent meta-analysis shows more frequently a receptor conversion for ER, PR, and HER-2, with most changes being from positive to negative receptor status [35]. In a multicenter study of 219 patients with BCBM, Hulsbergen et al., reported a receptor-specific disconcordance of 16.7% for estrogen, 25.2% for progesterone, and 10.4% for HER2. Initially HER2-negative patients gained HER2 in the BCBM by 14.8%. Loss of estrogen receptor was associated with worse survival [36]. As a tissue sample from the BCBM is not always accessible, there is a need for other approaches such as blood-based assays using circulation tumor cells (CTCs), cell-free tumor DNA, or microRNA.

### 1.3. Genomic Landscape of BCBM

Genomic analyses of BCBM and the corresponding primary tumor or other extracranial metastases have shown that BCBM may harbor potentially potent driver mutations that are not present in the corresponding primary breast tumor. The identification of brain metastasis-specific genomic alterations, with tailored development of targeted therapies directed toward identified proteins of these mutations, represents an important approach to improve the survival of patients with brain metastases.

One of the first and most important observations concerning the comparison of BMBC and primary breast tumors was the observation of differences regarding hormone receptor status. Retrospective studies have reported the loss of hormone receptor expression in brain metastases compared to corresponding primary breast tumors [37]. For example, using molecular profiling of paired brain metastases and corresponding primary breast tumors by whole-exome sequencing, Brastianos et al., showed that brain metastases exhibit genomic aberrations in the cyclin-dependent kinase (CDK) pathway and phosphatidylinositol 3-kinase/AKT/mammalian target of rapamycin (PI3K/AKT/mTOR) pathway; however, many of these alterations were not detected in the corresponding primary tumor [38].

In this context, it has also been postulated that the presence of an accumulation of replication errors is informative as an indicator of genomic instability. The human microsatellite loci hMLH1 (3p22, D3S1611), hMSH2 (2p16, D2S123), and NM23-H1 (17q21) were analyzed and correlated with the development of distant metastases in patients with early breast cancer. The phenotype of cells with a high replication defect correlated with the increased development of brain metastases, showing a relative risk ratio of 2.6 [39]. Several studies outlined an association between overexpression and a defect in the DNA repair pathway homologous recombination in brain metastases compared to the primary tumor. Woditschka et al., examined the gene expression profiles of 23 corresponding pairs of resected human brain metastases with their primary breast tumors. They observed overexpression of BRCA1-associated RING domain protein 1 (BARD1) and the recombinase RAD51 in BCBM compared with their corresponding primary tumors and with unrelated systemic metastases, respectively. In further analyses, they concluded that the overexpression of BARD1 and RAD51 may represent a mechanism to overcome reactive oxygen-species-mediated genotoxic stress in the metastatic brain [40]. Using a cohort of HER2+ BCBM, a gene expression signature that anticorrelates with BRCA1 overexpression was examined and termed BRCA1 Deficient-Like (BD-L) [41]. The evaluation of another independent cohort of BCBM showed significantly higher BD-L levels in BCBM compared to other metastatic sites. Although the BD-L signature was present in all breast cancer subtypes, it was significantly higher in BRCA1-mutated primary tumors than in sporadic breast tumors.

Ferguson et al., also studied individual pairs of primary tumor and brain metastases from non-small cell lung cancer, breast cancer, and melanoma. They observed that DNA topoisomerase II alpha (TOP2A) expression was elevated in brain metastases from all three cancers. Their further analysis showed that other proteins critical for DNA synthesis and repair, which may be associated with therapy resistance, were overexpressed in brain metastases. These included the ribonucleotide reductase catalytic subunit M1, thymidylate synthase, the DNA excision repair protein ERCC1, and topoisomerase I [42]. Additionally, the concordance of DNA copy number alterations (CNAs), mutations, and actionable genetic alterations (AGAs) was analyzed by comparative whole-genome array hybridization and targeted next-generation sequencing in primary breast cancer (PBC) and BCBM pairs [43]. They identified more CNAs, more mutations, and a higher tumor mutation burden, as well as more AGAs in BM than in PBC; 92% of pairs contained at least one AGA in BCBM that was not observed in paired PBC. This affected several therapeutically applicable inhibitor classes, including PI3K/AKT/mTOR inhibitors, poly(ADP-ribose) polymerase 1 (PARP) inhibitors, and CDK inhibitors. For PARP inhibitors, a defect in the DNA repair pathway homologous recombination was positive in 79% of BCBM compared with 43% of PBC. CDK inhibitors were associated with the largest percentage of discordant AGA that occurred in BCBM. Considering the AGA with the highest clinical level of evidence, 50% of pairs had an AGA in the BCBM that was not detected or considered in the analysis of paired PBCs. Thus, BCBM profiling provided a more reliable way than PBC to establish a diagnosis based on genomic analysis. Patients with BCBM deserve the investigation of various targeted therapies [43]. Yang 2020 analyzed the predictive value of DNA repair genes in postoperative metastasis of breast cancer. Protein expression of PARP1, X-ray cross complementing protein 4/1 (XRCC4, XRCC1), and ERCC1 were risk factors for postoperative metastasis of breast cancer. Postoperative metastasis of breast cancer could be effectively predicted when immunohistochemical scores involved PARP1 (IHC score) > 6, XRCC4 (IHC score) > 6, and ERCC1 (IHC score) > 3. Moreover, the combined analysis of PARP1, XRCC4, and ERCC1 had a large predictive value for the risk of BCBM.

Morgan et al., have summarized all relevant studies on this topic. The analysis of genetic alterations revealed mutations in order of decreasing frequency for the following genes: TP53, PIK3 subunit alpha (PIK3CA), lysine (K)-specific methyltransverase 2C (KMT2C), retinoblastoma gene 1 (RB1), zinc finger homeobox protein 3 (ZFHX3), BRCA2, HER2, lysine (K)-specific methyltransferase 2D (KMT2D), mismatch repair protein 1 (MLH1), phosphatase and tensin homolog (PTEN), ataxia telangiectasia and Rad3 related (ATR), BRCA1, epithelial cadherin 1 (CDH1), collagen alpha-3(VI) chain (COL6A3), FAT atypical cadherin 1 (FAT1), Fms related receptor tyrosine kinase 3 (FLT3), immunoglobulin-like and fibronectin type 1 (IGFN1), AT-rich interactive domain-containing protein 1A (ARID1A), ataxia telangiectasis mutated (ATM), checkpoint kinase 2 (CHEK2), mitogen-activated protein kinase kinase kinase 1 (MAP3K1), and MET proto-oncogene receptor tyrosine kinase (MET). In the context of BCBMs, which are frequently mutated, but not in extracranial metastases, genes involved in the cell-cycle, DNA repair pathways, and MET signaling were primarily identified [43]. In luminal BCBM, Cosgrove et al., identified a deficiency in homologous recombination (HDR) associated with mismatch repair defects. They hypothesize that understanding the relative contribution of specific mutational signatures in combination with RAD51 expression in BRCA1/2/PALB2 (partner and localizer of BRCA2) wildtype tumors may be important for predicting response to PARP inhibition in luminal BCBM, and they also call attention to the fact that functionally relevant HRD signatures exist in BCBM independent of somatic and germline BRCA1/2/PALB2 mutations [44].

In summary, significant differences in the genetic makeup of primary tumor and brain metastases were evident in these studies. Most of the abnormal genes were associated with cell-cycle and DNA repair mechanisms, with the DNA repair pathway Homologous Recombination being most frequently associated with differential expression in brain metastases.

## 2. Radiotherapeutic Treatment Strategies for BCBM

The treatment of brain metastases is complex, and a multidisciplinary approach should be discussed in future tumor conferences. In addition to radiotherapy, surgery and systemic drugs are available for the treatment of breast cancer brain metastases. Surgical resection should be considered for singular or solitary metastasis whenever possible, with an adjuvant local radiation [45,46,47]. In this context, singular only means a single metastasis in the brain, whereas a solitary brain metastasis is the only metastasis generally found. An improved survival benefit after surgery is shown for patients with a good KPS and a limited number of extracranial metastases [48,49]. The benefit of surgery is the immediate effect on symptomatic mass effects [50].

Radiotherapeutic treatment options are stereotactic radiotherapy/radiosurgery (SRT/SRS), fractionated stereotactic radiotherapy (FSRT), and whole-brain radiotherapy (WBRT) (Figure 1). WBRT for the treatment of brain metastasis was established in the 1950s [51]. For decades, WBRT was the gold standard in adjuvant treatment, as well as for unresectable brain metastases. It is usually delivered in a dose of 30 Gy in 10 fractions, with no benefit of lower or higher WBRT doses seen in a Cochrane Review [52]. Whole-brain RT is still a suitable option in palliation with multiple brain metastases, with prospective trials demonstrating complete or partial responses in approximately 60% of brain metastases treated with WBRT and the prevention of symptoms being apparent in about half of the patients [53]. Unfortunately, a major side effect is the impairment of the neurocognition. This seems to be related to a loss of neuronal stem cells in the hippocampal area, but the area itself is rarely affected by metastases [54]. There are some data demonstrating an improved neurocognitive outcome after hippocampal-avoidance WBRT, albeit with no difference in intracranial progression-free survival and overall survival. This option should be considered for patients undergoing WBRT with a good performance status and prognosis [55,56]. WBRT in combination with resection or radiosurgery improves progression-free survival but not overall survival, and should therefore only be considered as an individual option [57,58,59].

The SRS was first established for malignant brain lesions and other neurosurgical disorders where the head of the patients was immobilized by invasive techniques to enable a very precise high-dose delivery. Interestingly, the first stereotactic radio surgical apparatus was also presented in the 1950s. With innovations in imaging, radiobiological insights, and non-invasive immobilization techniques, this concept was further developed to FSRT and extracranial targets (stereotactic body radiotherapy), and differentiation between SRS and SRT became less well defined [60].

The single fraction approach with high doses up to 24 Gy has the advantage of a short treatment duration, but there is a trend towards hypofractionated regimes of 3–5 therapy sessions, which offer the benefit of a higher therapeutic ratio. The optimal regime for brain metastasis regarding local control, but also including side effects such as radionecrosis, is still under investigation [61,62,63]; see Alliance trial NCT04114981. SRS/SRT is an alternative to surgery for smaller lesions (<3 cm) and is recommended for a limited number of metastases with a treatable volume. It has a favorable impact on neurocognition compared to whole-brain radiation and receives good local control rates. Yamamoto et al., demonstrated the non-inferiority of SRS for 5–10 BM compared to SRS of 2–4 BM [64]. Postoperative SRT/FSRT is an established adjuvant treatment that reduces the risk of local failure and shows superior preservation of neurocognition when compared to adjuvant WBRT (NCCTG N107C/CEC3), [53,65,66].

Moreover, 3D-conformal photon radiotherapy delivered by a linear accelerator is the most frequently used state-of-the-art technique, and it includes intensity-modulated radiotherapy, where a photon beam is shaped dynamically by multileaf collimators (≤5 mm) to create a complex dose distribution and a multi-field/-arc SRT with a steep dose fall off toward surrounding tissue. Accuracy is enhanced by in-room imaging and is often supported by positioning systems using image fusion algorithms.

To date, RT guidelines for BM have been generated from studies including multiple primary tumor entities, so a more individualized approach is highly warranted (Figure 2). Patients with hormone-receptor-positive and/or HER2-positive BCBM are estimated to have a longer survival after SRT and show the greatest benefit [67]. This leads to the question of how quality of life can be preserved best in this group and whether RT ought to be escalated in the TNBC subgroup.

### 2.1. Radioresistance in BCBM

Despite the advantages of RT in MRI-based planning and the sub-millimeter precision of modern BM irradiation, median survival of patients with BCBM is 16 months, and local relapse occurs frequently [31]. The curative potential of RT in breast cancer treatment is clear not only in the adjuvant setting after resection of the primary tumor but also in the case of axillary lymph node metastases detected by a positive sentinel node biopsy, where axillary radiotherapy is non-inferior to axillary lymph node dissection [69,70,71].

When discussing reasons for therapy failure, the definition of CSC, i.e., a cancer cell with the potential to self-renew and generate a heterogeneous lineage of cells, leads to the hypothesis that radiotherapy is only successful when none of these cells are left after irradiation to cause recurrence. This and this topic will be discussed in greater depth below. Besides the number of CSCs, multiple other radiobiological factors such as tumor hypoxia, along with reoxygenation and repopulation capacity and DNA repair efficacy, have been extensively studied in experimental and clinical settings [72].

Despite numerous efforts in this field, there are no clinically approved predictors of radioresistance for BCBM or for breast cancer in general. Eschrich and colleagues validated a 10-gene expression radiosensitivity signature as a predictive biomarker of RT benefits in breast cancer [73], while Yan and colleagues suggested a combined model integrating immune- and hypoxia-related gene signatures integrating microenvironment-related factors for improved prediction [74]. Recently, Monteiro et al., found that activation of the S100A9–RAGE–NF-κB–JunB pathway in brain metastases mediates resistance to WBRT. They further identified that S100A9 expression correlates with clinical response to WBRT for BM (including breast and lung adenocarcinoma, melanoma), indicating its potential as a noninvasive biomarker [75].

In addition to these efforts to generate a serum-based biomarker, radiomics are another promising attempt to predict therapy response to RT. It is assumed that small variations in pixel/voxel intensity, density, and position, analyzed by applying artificial intelligence, can serve as biomarkers for patient stratification. With easily accessible MRI scans, radiomics are a powerful tool to personalize the management of BCBM, and they have the potential to predict responses to RT, tumor mutation status, and discriminate recurrence from radiation necrosis [76].

### 2.2. Systemic Therapies of BCBM and RT

In the complex setting of BCBM, systemic therapies, especially molecular-targeted therapies such as a HER2-directed therapy, are often indicated for systemic control. The relation of systemic therapies and concomitant RT remains imprecise. Clinical trials often exclude patients with brain metastasis, leading to a small body of evidence when it comes to CNS effectiveness and side effects, with even fewer data on concurrent radiotherapy. Considering the limited effectiveness of the available treatments, synergistic effects of RT and systemic targeted therapies are highly desired, and the radiosensitization of resistant cells such as CSC is urgently needed. With the limited prognosis of patients, quality of life and therapy-related morbidity also require greater focus.

Although hormone-receptor-positive breast cancer is not the subtype most likely to metastasize to the brain, there are several anti-hormonal therapies available for ER and/or PR-positive metastatic breast cancer. Despite reports of a high level of tamoxifen in the brain and some case reports of a clinical benefit, the effects of this estrogen receptor modulator or other hormone therapies on brain metastases are elusive [77]. Interestingly, there are some data that suggest a potential benefit of tamoxifen in the BCBM of estrogen-receptor-deficient breast tumors by modulating the microglia and increasing their antitumor phagocytic ability. Furthermore, estrogen-stimulated microglia were shown to promote tumor stem cell growth by secreting CCL5, indicating that blocking this pathway might be another essential benefit of tamoxifen [78]. It is also reported that tamoxifen modulates the drug resistance of BCBM through a decrease in interleukin 6 (IL 6) expression in astrocytes, as well as through the downregulation of mitogen-activated protein kinase 1 (MAPK), Janus kinase 2 signal transducer, and the activator of transcription 3 signaling pathway (JAK2/STAT3) in hormone-receptor-negative cancer cells [79].

The combination of cyclin-dependent kinase 4 and 6 (CDK4/6) inhibitors with hormone receptor antagonists is an established treatment strategy, as the activation of the CDK 4/6 pathway is a well-known resistance mechanism. Interestingly, there are some preclinical data showing antitumor activity in both ER-positive breast cancer cell lines and ER-negative cell lines; this could be related to the inhibitory effects on RB1 phosphorylation, G1-S cell-cycle progression, cell senescence, and the proportion of CSCs. The authors concluded that CDK 4 inhibition may cause anti-CSC activity [80]. Pre-clinical data comparing CDK 4/6 inhibitors confirmed that ademaciclib crosses the BBB more efficiently than palbociclib [81]. Data outlining the synergistic efficacy of CDK 4/6 inhibitors and irradiation were mostly generated from glioblastoma models [82,83]. Figura et al., published a small retrospective series of SRT and CDK 4/6 inhibition in BCBM, concluding that the combination was well tolerated and, compared to historical data, brain metastases control rates are similar, whereas survival seemed prolonged [84]. There are no prospective trials with CDK 4/6 inhibition combined with RT in BCBM, but phase II trials evaluating the role of ademaciclib and palbociclib without RT—both of which show a good safety profile with a rather low response rate—highlight the need for therapy improvement through additional local ablative therapy (NCT02308020), (NCT02774681).

Inhibition of HER2 in metastatic breast cancer with an HER2 amplification is an established treatment option and improved the outcome of this subgroup. HER2 overexpression is correlated with radioresistance [85,86,87]. One possible underlying mechanism is an activation of NF-κB upon irradiation, inducing further HER2 overexpression and leading to radioresistance in a positive feedback loop [88]. Notably, in some studies it has been shown that irradiation induced HER2 expression is associated with a cancer stem cell phenotype [27,86]. Furthermore, several studies revealed that HER2 overexpression/activation is a key regulator of EMT and CSC cell programs in HER2-negative and HER2-positive BC [89,90,91].

There is good preclinical evidence for radiosensitization by HER2 inhibition [92,93,94], and clinical data have not indicated increased toxicity in general [95]. The first approved HER2 inhibitor, trastuzumab, was found to be relatively ineffective in preventing brain metastasis, which is assumed to be due to its heavy molecular weight and therefore a poor penetration of the BBB. Lapatinib, a dual tyrosine-kinase inhibitor (TKI) of EGFR and HER2, is a small molecule, but a direct comparison of trastuzumab/capecitabine and lapatinib/capecitabine failed to prove a better prevention of BCBM under TKI treatment (CEREBEL trial, [96]), with modest intracranial activity in monotherapy [97] and increased responses rates in combination with capecitabine [98]. There are more data showing an improved local control rate after SRS with concurrent HER2/EGFR TKIs supporting this approach for patients without extracranial disease [99,100,101,102]. Moreover, Khan et al., performed a meta-analysis and concluded that lapatinib has good intracranial activity and achieves better survival for patients with HER2 positive BCBM; simultaneous SRT was associated with better local control and survival [103].

Radionecrosis is one of the main complications of CNS RT and is associated with significant morbidity [104]. An increased incidence of STR-induced radionecrosis in HER2-positive BCBM was reported after adjuvant exposure to trastuzumab emtansine, an HER2-antibody–drug conjugate [105,106]; however, this could not be reproduced in a series by Mills and colleagues [107]. In another retrospective study, Park et al., found a significantly higher proportion of patients developing radionecrosis in the group receiving more than one HER2-directed agent, but no significant difference was apparent when comparing patients with one or no HER2-directed agent during SRT [108].

There are several other HER2-directed therapies such as pertuzumab, trastuzumab-deruxtecan, neratinib, and tucatinib, the latter of which recently demonstrated improved antitumor activity against BCBM in a randomized controlled trial, and therefore will probably become an essential part of the treatment in HER2-positive metastatic breast cancer [109]. Unfortunately, to date, there are no data available regarding concurrent radiotherapy, but the promising results of tucatinib indicate the need for further investigations with concurrent irradiation.

The PI3K/AKT/mTOR pathway is a common target in cancer therapy, as it is a key regulator of cell proliferation and metabolism, and over-activation of this pathway is associated with tumor development, progression, and drug resistance. The FDA and EMA approved the PI3K inhibitor alpelisib and the mTOR inhibitor everolimus for the treatment of advanced ER+ breast cancer [110]. While there is preclinical evidence supporting their potential radiosensitizing effect, clinical data are limited, with no breast-cancer-specific studies [95]. Of note, PIK3 gene alterations were found to be the second-most-mutated gene in BCBM [111].

A direct way to achieve radiosensitization is the targeting of proteins involved in the DNA damage response, as irradiation leads to DNA lesions. PARP is involved in single- but also double-strand break repair. In the presence of deficiencies in homologous recombination repair, such as that caused by a BRCA mutation, PARP inhibition is an effective option and has been approved for the treatment of metastatic breast cancer with a germline BRCA mutation [112]. Veliparib is a PARP inhibitor that can cross the BBB [113]. Mehta et al., performed a phase I trial combining veliparib with WBRT in brain metastasis of solid tumors (breast cancer n = 25); it showed encouraging safety and preliminary efficacy results [114]. Taken together, there are some results pointing towards a potential benefit when combining radiotherapy with these molecular-targeted agents, but with almost no clinical data available on safety and efficiency, their future role in the treatment and radiosensitization of BCBM remains elusive.

Solid evidence for the use of chemotherapy agents for BCBM is also limited, but historic trials reported a response rate of >30% for various agents (e.g., capecitabine, 5-FU, MTX, etoposide, cisplatin, vincristine). Therefore, these non-targeted systemic therapies are proposed for use in HER2-negative patients [115].

## 3. Mechanisms of Radioresistance and Immune Evasion in CSCs

The concept of CSCs was first established in acute myeloid leukemia. A small fraction of cells showed the ability to engraft a new host, while others, so-called bulk cells, failed to engraft [116]. Subsequent studies identified CSC activity in breast cancer (BCSC) and other solid tumors. BCSCs were identified by the expression of the cell surface markers CD44(+)/CD24(−/low) using cell-sorting and xenografting approaches [117]. Despite ongoing intensive research on this unique tumor cell subpopulation, the definition and identification of tumor-initiating cells remain elusive. However, it is becoming gradually clear that CSCs (comparable to tissue stem cells) need not be rare and/or quiescent. Many examples demonstrate that they can be abundant and proliferate extensively. Moreover, it is becoming clear that stem cell hierarchies can be much more plastic than previously thought, a phenomenon that complicates the identification and eradication of CSCs.

The original model of a strict cellular hierarchy, starting from the CSC down to the differentiated somatic cell, is currently being considered [44]. It has been observed that the potential of putatively fixed cells to move up and down the hierarchy of differentiation (“plasticity”) is widespread. Several studies showed that both CSCs and non-CSCs are plastic and undergo phenotypic changes in response to appropriate stimuli. The plasticity of non-CSCs is influenced by the microenvironment, both in the primary tumor and in the metastatic situation. Epithelial cancer cells can acquire a mesenchymal gene program that facilitates migration and invasion. This process is referred to as epithelial-to-mesenchymal transition (EMT). In recent years, the relationship between cellular stem cells and EMT has attracted considerable attention. It is known that overexpression of EMT transcription factors not only enhances a mesenchymal–migratory phenotype but also increases the tumor-initiating potential of cell lines. Tumor cells with increased endogenous levels of Snail Family Transcriptional Repressor 1 (SNAI1), the major EMT transcription factor, exhibit an increased tumor-initiating ability and metastatic potential [118].

Interestingly, breast cancer xenografts showed that migrating cells that had undergone EMT returned to the epithelial state immediately after reaching the metastatic site. This contradicts the hypothesis that EMT is necessary for phenotype maintenance and suggests that EMT may be uncoupled from stem cells in defined contexts. EMT could be transient in cancer cells and, depending on environmental factors, adopt an intermediate mesenchymal state that is reversible. These transitions result in a plastic CSC phenotype. Supporting this, human basal breast cancer cells switch between non-CSC and CSC states depending on the expression of the EMT inducer Zinc Finger E-Box Binding Homeobox 1 (ZEB1). To this end, in non-CSCs, the ZEB1 promoter is maintained in a bivalent chromatin configuration that allows cells to respond rapidly to EMT-inducing signals from the microenvironment and, consequently, increase their tumorigenic capacity [119]. Taken together, these and other studies suggest that CSC hierarchies are not rigid in many cancers [120].

Importantly, most observations regarding the radioresistance of CSCs relate to the results of studies on CSCs of primary tumors. Therefore, it is assumed that the observations in BCBM are comparable to those of glioblastomas. This may not correspond to reality because, unlike glioblastomas, BCBMs must have the additional potential to escape the primary tumor, migrate through the bloodstream, invade the neural niche, and initiate new tumor growth in a physiologically altered secondary niche. To our knowledge, there is only one study addressing this question. In this study, new animal models were established to investigate early tumor adaptation in brain metastases. For this, mice were administered both patient-derived and cell line-derived CSC-enriched tumor sphere cells from brain metastases. Astrocytes were observed to affect the activation of pro-cadherin 7 (PCDH7)-PLCb-Ca2þ-CaMKII/S100A4 signaling as a mediator of tumor growth in brain metastases. [121].

### 3.1. Cellular Processes Leading to Radioresistance in CSC

Various cellular processes lead to the radioresistance of CSC. These include a low level of reactive oxygen species (ROS). ROS are involved in various physiological processes such as proliferation, differentiation, metabolism, apoptosis, angiogenesis, wound healing, and motility [122]. Intracellular ROS levels are tightly and continuously regulated by ROS scavengers such as superoxide dismutase, superoxide reductase, catalase, glutathione peroxidase, glutathione reductase, or apurin/apyrimidine endonuclease/redox effector factor (Ape1/Ref-1, also known as APEX1). ROS scavengers appear to be upregulated and highly efficient in the CSCs of various tumors, resulting in low ROS levels and protecting CSCs from RT-induced cell death. In this sense, CSCs have been shown to produce fewer ROS upon irradiation than non-CSCs [122]. Among the accumulating evidence of radiation-resistant mechanisms in CSCs, such as altered cell-cycle distribution toward more resistant phases, tumor cell repopulation, and hypoxia [123], the most widely accepted concepts are increased DNA damage repair capacity to eliminate DNA double-strand breaks after IR [124,125,126].

### 3.2. DNA Repair Mechanisms Contributing to Radioresistance in CSC

It was observed that breast cancer-initiating cells are resistant and increase in number due to the loss of bulk cells after IR [127,128]. Landmark studies have shown that radioresistance directly correlates with the number of CSCs. This is mainly attributed to an upregulated DNA damage response and DNA repair capacity. The contribution of DNA repair complex non-homologous end-joining (cNHEJ) to radioresistance has not been conclusively identified. There is considerable evidence for the importance of DNA repair by homologous recombination (HR), which is activated by the intra-S phase checkpoint kinase (CHK1) determining the radioresistance of BCSCs. In accordance, increased expression of RAD51 was detected in aldehyde dehydrogenase 1 family member A1-positive breast cancer cells. In addition, the multifunctional DNA repair protein BRCA1 appears to regulate the level of BCSC-like populations through epigenetic changes (summarized in [125,126]).

Kim et al., demonstrated that BRCA1 plays a critical role in regulating CSC-like traits. While downregulation of BRCA1 resulted in a significant increase in CSC-like populations, a significant decrease in CSC-like populations was observed in breast cancer cells after reconstitution of BRCA1. Moreover, BRCA1-reconstituted tumor cells are more sensitive to histone deacetylase (HDAC)-inhibitor-induced loss of stem cell function corresponding to a BRCA1-deficient phenotype. Surprisingly, hypoxia preferentially blocked HDAC inhibitor-induced differentiation of BRCA1-reconstituted breast cancer cells [129].

The mechanisms underlying glioma stem cell (GSC) radioresistance may be that chromatin state and DNA lesion accessibility to DNA repair mechanisms are critical for maintaining genomic stability. Obara et al., presented results from a high-content, small-interfering RNA microscopic screen showing that the SPT6 transcriptional elongation factor is critical for genomic stability and self-renewal of GSCs. Mechanistically, SPT6 upregulates the transcription of BRCA1, thus driving error-free DNA repair in GSCs. Loss of SPT6 impairs self-renewal, genomic stability, and tumor-initiating ability of GSCs [130].

Confirming this, Lim et al., (2012) observed that differentially radiosensitive glioblastoma cells did not exhibit altered regulation of cNHEJ. Indeed, cNHEJ was equivalent or reduced in glioma-initiating cells compared to non-tumor-forming neural progenitor cells. There was, however, evidence of more efficient repair of homologous recombination in glioma-initiating cells. It was observed that neither prolonged cell cycle nor enhanced basal activation of checkpoint proteins occurred. Instead, cell-cycle defects were observed at G1 S- and S-phase checkpoints by controlling S-phase entry and radioresistant DNA synthesis after irradiation. These data suggest that homologous recombination and cell-cycle checkpoint abnormalities contribute to the radioresistance of glioma-initiating cells [131]. Supporting these observations, Chen et al., identified the DNA repair suppressor Leucine-Rich Repeat-Containing Factor 31 (LRRC31) via a genome-wide CRISPR screen. LRRC31 interacts at the protein level with Ku70/Ku80, and ataxia telangiectasia mutated and RAD3-related protein (ATR). Overexpression of LRRC31 suppresses DNA repair and sensitizes BCBMs to radiation. They also showed that targeted delivery of the LRRC31 gene via nanoparticles improved survival of tumor-bearing mice after irradiation [132].

Taken together, most of the data suggest that the interplay of predominant DNA repair and balanced regulation of the DNA damage response in S-phase allows CSCs a significant survival advantage after irradiation.

### 3.3. Immune Evasion of CSCs in BCBM

In recent years, remarkable progress has been made in the field of cancer immunology, and many of the mechanisms by which tumors prevent elimination by the immune system are now understood. However, the initial events of immune evasion have not yet been conclusively elucidated. Recent studies showed that CSCs can hide from the immune system and evade the immune surveillance phase. Agudo et al., showed that the immune privilege of epithelial stem cells is also related to their proliferative state and is not an inherent property that they possess permanently. They observed that circulating epithelial stem cells are eliminated by the immune system, whereas slowly cycling stem cells escape immune recognition. This escape is the result of a systematic downregulation of antigen presentation machinery, rendering stem cells virtually invisible to the adaptive immune system. Increasing the expression of the transcriptional transactivator Nlrc5, which is not expressed in the resting state, restored major histocompatibility complex class I (MHC-I) expression on stem cells. These studies demonstrate that some tissue stem cells hide from immune surveillance and protect their integrity [133]. Mechanisms of immune evasion were also observed in BCSC. ALDH1-positive BCSC showed decreased expression of the antigen-processing gene-associated transporter (TAP), the co-stimulatory molecule CD80, and programmed cell death 1 ligand (PD-L1) [134]. Hsu et al., observed that EMT enriches PD-L1 in CSCs through the EMT/β-catenin/STT3/PD-L1 signaling axis, in which EMT transcriptionally induces the N-glycosyltransferase STT3 through β-catenin, and subsequent STT3-dependent PD-L1 N-glycosylation stabilizes and upregulates PD-L1. This axis is also used by the non-CSC population but has a much more profound effect on CSCs, as EMT induces more STT3 in CSCs than in non-CSCs [135].

DNA damage arising in S phase also has a stimulatory effect, not only directly on PD-L1 expression, but also on intracellular immune activation through the appearance of cytosolic DNA. Both processes are dependent on the activation of the ATR-CHK1 pathway. Thus, there appears to be a direct link between the DNA damage response and immune evasion triggered by HR-mediated processes and activation of the DNA damage response in the S phase. These observations also suggest that the innate immune response, especially in BCSCs, should be harnessed by inhibiting their effective DNA repair mechanisms to successfully employ novel therapeutic approaches [131]. Moreover, Sato et al., observed that PD-L1 expression is upregulated in cancer cells in response to DNA damage. This upregulation requires ATM/ATR/CHK1 kinases and is further amplified by deletion of BRCA2 followed by enhanced CHK1-dependent upregulation of PD-L1 after IR or PARP inhibition. The generated DSBs activate STAT1 and STAT3 signaling, as well as IRF1, which is required for DSB-dependent PD-L1 upregulation. These results demonstrate that DSB repair is involved in PD-L1 expression, and they shed light on how PD-L1 expression is regulated after the induction of DSBs [136].

## 4. New Strategies to Target Radiation Resistance in CSCs

### 4.1. Targeting DNA Repair in BCBM

As described previously, the high radioresistance of BCBM and glioblastomas is attributed, in part, to the presence of CSCs that promote both G2/M checkpoint activation and efficient DNA repair. Novel treatments to enhance IR efficacy have focused on targets within DNA damage response (DDR) signaling pathways (Figure 3).

Balbous et al., analyzed the radiosensitizing effect of inhibiting RAD51. They observed that GSC exhibited significantly more DNA damage and decreased survival after irradiation and concurrent treatment with a RAD51 inhibitor. The authors conclude that inhibition of RAD51 may be an evolved therapeutic strategy [137].

Antonelli et al., reported the efficacy of inhibiting ATM. They demonstrated that ATM acts as a tumor-promoting factor in HER2-positive breast cancer and that ATM expression maintains the proportion of cells with a stem-like phenotype independent of HER2 expression level. In this regard, ATM in mammospheres modulates the expression of genes related to the cell cycle, DNA repair, and autophagy. Knockdown of the autophagy gene, autophagy-related 4C-cysteine peptidase (ATG4C), impairs mammosphere formation in a manner comparable to ATM depletion. Conversely, ectopic expression of ATG4C in cells in which ATM expression is knocked down restores mammosphere growth. Likewise, they observed a significant correlation between ATM and ATG4C expression levels in all human breast cancer subtypes except the basal type. They concluded that ATM and ATG4C in breast cancer cells are essential drivers of mammosphere formation, suggesting that targeting them may improve current approaches to eradicate breast cancer cells with a stem-like phenotype [138].

Liu et al., (2017) investigated the effect of PARP inhibitors in BCBM of TNBC. They demonstrated that CSCs in BRCA1-mutated TNBCs were resistant to PARP inhibition and that these cells exhibited both increased RAD51 protein levels and activity. Downregulation of RAD51 by shRNA sensitized CSCs to PARP inhibition and reduced tumor growth. BRCA1 wildtype cells were relatively resistant to PARP inhibition alone, but the reduction in RAD51 sensitized both CSCs and bulk cells in these tumors to PARPi treatment. The authors interpreted their data to suggest that both BRCA1-mutated and BRCA1-wildtype TNBC CSCs are relatively resistant to PARP inhibition [139].

In 2020, Zenke et al., investigated the radiation-sensitizing effect of the DNA-dependent protein kinase catalytic subunit (DNA-PKcs) inhibitor, M3814 (peposertib). M3814 effectively inhibits the catalytic activity of DNA-PKcs and sensitizes several cancer cell lines to ionizing radiation and DSB-inducing agents. Inhibition of DNA-PKcs autophosphorylation in cancer cells or xenograft tumors resulted in increased numbers of persistent DSBs. Oral administration of M3814 to two xenograft models of human cancer using a clinically established 6-week fractionated irradiation regimen potentiated the antitumor activity of IR and resulted in complete tumor regression at nontoxic doses. These results argue for DNA-PKcs inhibition as a new approach for combined radiation therapy of cancer, and M3814 is currently being investigated in combination with radiotherapy in clinical trials [140].

Mampere et al., postulated that so-called long noncoding RNAs and transcription factors targeting the metastasis-associated lung adenocarcinoma transcript, HOX transcript antisense RN, long-coding RNA breast metastasis, triglyceride lipase-cholesterol esterase 1, and activating transcription factor 3 genes have the potential to both prevent metastatic spread and treat BCBM with increased radiosensitivity. Given the propensity of HER2+ breast cancer to develop into BCBM, these cell lines may be an important target for future investigation [141].

Liu et al., reported that brain-enriching long noncoding RNA (BMOR) expressed in BCBM cells is required for the development of BCBM to induce cancer cells to colonize brain tissue. Mechanistically, BMOR enables cancer cells to evade immune-mediated killing in the brain microenvironment, allowing BM to develop by binding and inactivating IRF3. In preclinical mouse models with BCBM, they demonstrated that a silencer targeting BMOR is effective in suppressing the metastatic colonization of cancer cells in the brain [142].

### 4.2. Targeting DNA Repair and Immune Response in BCBM

Currently, increasing preclinical studies are being conducted on the combination of DDR inhibitors and immune checkpoint inhibitors (ICI) with irradiation in various tumor models (Figure 3). Experimental approaches in BCSCs and BCBMs are pending.

Vendetti et al., (2018) showed that the ATR kinase inhibitor AZD6738 attenuated radiation-induced CD8+ T cell depletion and enhanced CD8+ T cell activity in mouse models of Kras-mutated cancer in combination with conformal radiotherapy. Mechanistically, AZD6738 blocks radiation-induced PD-L1 upregulation on tumor cells and dramatically reduces the number of tumor-infiltrating regulatory T cells (Tregs). Of note, AZD6738, in combination with conformal radiotherapy, was able to generate immunological memory in mice with a complete response [143].

Additionally, Dillon et al., (2019) investigated the effects of ATR inhibition by AZD6738 in combination with fractionated RT in an immunocompetent mouse model of HPV-related malignancies. They showed significant radiosensitization after IR by ATRi, in addition to more DNA damage and a marked increase in immune cell infiltration. They noted increased numbers of CD3þ and NK cells. ATR inhibition plus IR resulted in a gene expression signature consistent with a type I/II interferon (IFN) response. Increased MHC I levels were observed on tumor cells, with transcriptional level data indicating increased antigen processing and presentation in the tumor. In vivo, significant modulation of cytokine gene expression (particularly CCL2 and CCL5, and C-X-C motif chemokine ligand 10) was observed, with in vitro data indicating that CCL2, CCL5, and CXCL10 were produced by tumor cells after ATR inhibition plus RT [144].

Sheng et al., (2020) investigated the role of the ATR inhibitor AZD6738 on the combination of IR and ICI in hepatocellular carcinoma. AZD6738 was found to increase IR-stimulated CD8+ T-cell infiltration and reverse the immunosuppressive effect of IR on the number of Tregs in mouse xenografts. Moreover, the addition of AZD6738 enhanced infiltration, increased cell proliferation and the ability to produce IFN-γ from tumor-infiltrating lymphocyte (TIL) CD8+ T cells, caused a decreasing trend in the number of TIL, Tregs, and depleted T cells in mouse xenografts. Thus, the tumor immune microenvironment was significantly improved [145].

In an HPV-negative murine mouse model of oral squamous cell carcinoma, Patin et al., (2022) observed that inhibition of ATR enhances IR-induced inflammation of the tumor microenvironment, with natural killer (NK) cells playing a central role in maximizing treatment efficacy. They observed that ICI can further enhance the antitumor activity of NK cells [146].

In pancreatic cancer, Zhang et al., (2019) postulated that activation of the innate immune response through enhanced induction of DNA damage could further increase the efficacy of ICI. They showed that inhibition of ATM alone was able to increase tumoral type I IFN expression independently of the cyclic GMP-AMP synthase (cGAS) and the stimulator of the interferon response CGAMP interactor 1 (STING), but this was only achievable in a manner dependent on tank-binding kinase 1 (TBK1) and the proto-oncogenic tyrosine protein kinase Src. The combination of ATM inhibition and IR increased TBK1 activity even more markedly and, correspondingly, also increased IFN production and antigen presentation. In addition, the silencing of ATM increased PD-L1 expression and increased the sensitivity of pancreatic tumors to PD-L1-blocking antibodies. This was associated with an increase in tumoral CD8+ T cells and established immune memory. The authors further observed that low ATM expression inversely correlated with PD-L1 expression in pancreatic tumors from patients. Overall, these results demonstrate that the efficacy of ICI in pancreatic cancer is enhanced by ATM inhibition and further potentiated by irradiation, depending on the increased immunogenicity of the tumor [147].

Wang et al., (2022) examined whether DNA-PKcs, the kinase critical for repair via cNHEJ in cancer cells, is immunomodulatory. They observed that the combination of IR and DNA-PKcs inhibition induces cytosolic double-stranded DNA and tumoral type 1 IFN signaling independently of cGAS and STING. That said, DNA-PKcs inhibition and IR also promote PD-L1 expression. The use of anti-PD-L1 in combination with IR and DNA-PKcs inhibitors potentiates antitumor immunity in pancreatic cancer models [148].

Patel et al., (2019) investigated the importance of inhibition of the G2 checkpoint kinase 1 (Wee1) in combination with IR on killing by T lymphocytes and the sensitizing effect for ICI. In several models, they observed that the Wee1 inhibitor AZD1775 induced DNA damage accumulation. The combination treatment improved the control of a syngeneic mouse model of oral cancer (MOC1) in vivo, and the on-target effects of systemic AZD1775 were localized with targeted IR. The combination treatment enhanced granzyme B-dependent T lymphocyte killing by reversing the additive G2/M cell-cycle blockade induced by IR and granzyme B. The combination of IR and AZ1775 improved CD8+ cell-dependent control of MOC1 tumor growth and the rate of complete rejection of established tumors in the context of PD axis ICI. Functional assays demonstrated enhanced tumor anti-gene-specific immune responses in sorted T lymphocytes. The combination of IR and AZD1775 not only increased tumor-specific cytotoxicity but also improved susceptibility to killing by T lymphocytes and response to PD axis ICI [149].

In summary, all recently tested inhibitors of the DNA damage response in combination with IR have resulted in enhanced immunomodulation.

### 4.3. Targeting Immune Checkpoints in BCBM

Cancer immunotherapy has changed the therapy landscape and thus the prognosis of many tumor entities, e.g., malignant melanoma and lung cancer. Neutralizing antibodies targeting PD-1 receptor of lymphocytes, its ligand PD-L1, and the cytotoxic T-lymphocyte-associated protein 4 (CTLA-4) “unleash” the tumor-specific T-cell response [150] (Figure 3).

Pembrolizumab is already an established first-line treatment for metastatic or recurrent unresectable TNBC in combination with chemotherapy, and it was recently approved by the European Commission in the neoadjuvant setting for early stage or locally advanced TNBC [151]. The impact of radiotherapy on the immune response is controversial, though there is growing evidence of the ability of RT to induce immunogenic cell death and modify the tumor microenvironment. This has led to an evolving number of clinical trials combining immune checkpoint inhibitors (ICIs) with radiotherapy in breast cancer; however, data on BCBM are limited.

A subgroup analysis of the IMpassion130 trial investigating atezolizumab and nab-paclitaxel for TNBC failed to prove a benefit on BCBM; notably, this analysis was underpowered due to the small subpopulation. Two large meta-analyses with brain metastasis from non-small-cell lung cancer and malignant melanoma indicated prolonged overall survival with simultaneous immunotherapy and radiosurgery [152,153]. A nonrandomized phase 1b trial of nivolumab one week before SRS and every four weeks after demonstrated some activity in certain BCBMs and good safety in the preliminary data (NCT03807765) [154]. CTLA4 inhibitor tremelimumab (± HER2-directed therapy with trastuzumab) and concurrent RT were investigated in 26 patients and showed modest clinical activity in the HER2-negative cohort and encouraging responses in the HER2-positive cohort (NCT02563925) [155].

At present, there is also a phase II clinical trial studying the combination of atezolizumab in TNBC brain metastasis (NCT03483012), as well as a clinical phase I/II trial (NCT03449238) combining the SRS of selected BCBMs with pembrolizumab, the latter of which was designed to investigate the effect of this combined treatment on other untreated CNS metastases.

Regarding side effects, a pooled analysis including 68 prospective trials of patients receiving immunotherapy found no association of an increased risk of serious adverse events when ICIs were administered within 90 days after RT [156]. As mentioned above, radionecrosis after CNS RT is a side effect of particular interest. There are limited data regarding the risk of radionecrosis, but existing evidence suggests acceptable safety when combining immunotherapy and RT of brain metastases [157,158]. However, data generated from melanoma patients receiving SRS and CTLA-4 blockade indicate a higher radionecrosis risk [159], with no breast-cancer-specific evidence available.

The abscopal effect describes a phenomenon in which local radiation triggers a systemic antitumor effect. In the era of conventional fractionated radiotherapy, this was the chief type of anecdotally reported reaction, and published case reports mostly refer to immunogenic cancer, such as malignant melanoma. It is assumed that cell death by IR leads to a release of immunogenic factors and different endogenous-damage-associated molecular patterns (DAMPs), which themselves activate dendritic cells and antigen presentation to T cells. On the other hand, there are also immunosuppressive effects of IR that are mediated by increased TGF-β levels, the attraction of regulatory T cells, and myeloid-derived suppressor cells. Almost two decades ago, preclinical results from a model with poor immunogenic metastatic mouse mammary carcinoma demonstrated an increased survival when the CTLA-4 blockade was combined with RT, which correlated with the inhibition of lung metastases formation [160]. Recently, there has been growing evidence of the crucial role of the immune response after treatment, even for tumor entities considered to be less immunogenic such as breast cancer.

Historically, the role of the CNS immune system on the systemic immune response in solid tumors has been underrated, but evidence on infiltrates of lymphocytes and leukocytes, which seem to be even more prominent compared to primary brain tumors [161], has attracted interest. In a systematic review of the abscopal effect in patients with brain metastasis, Pangal et al., found that most cases present an extra-cranial response to intracranial irradiation; there was also one report of an intracranial response after axilla irradiation, leading the authors to conclude that that the immune response is enabled by a disrupted BBB thanks to metastases and RT itself [162].

There is an ongoing debate on the optimal dose and fraction of RT combined with immunomodulation, with no established regime. There are some data showing hypofractionated RT to be more effective in initiating an immune response than high-dose single fractions.

It is suggested that the cancer-cell-intrinsic activation of the type I IFN pathway and the production of IFNβ are essential mechanisms of antitumor T cell generation by radiation, and therefore essential mediators of abscopal responses. The DNA three-prime repair exonuclease Trex1 (TREX1) is estimated to have a diminishing effect on immunogenicity by degrading accumulated DNA in the cytosol; this could activate the cGAS-STING pathway, which essential for CD8+ T cells that mediate systemic tumor rejection in the context of ICI. TREX1 is induced by radiation doses above 12–18 Gy in different cancer cells and attenuates their immunogenicity by degrading DNA that accumulates in the cytosol upon radiation [163].

In another breast cancer mouse model, Dewan and colleagues achieved an enhanced tumor response at the primary site, as well as significant growth inhibition of tumors outside the radiation field with fractionated regimes (3 × 8 Gy or 6 × 5 Gy), but not after 20 Gy in a single fraction. In addition, they reported that tumor-specific IFN-gamma production by CD8+ T cells correlated with this abscopal effect [164]. Similarly, to these results, Morrisa et al., showed a synergistic primary and distant tumor control combining PD-1 blockade with hypofractionated RT (2 × 8 Gy), but not after low-dose irradiation (10 × 2 Gy) [165]. It is suggested that the high level of tissue damage after high single-dose RT (e.g., 1 × 15 Gy or more) might induce an anti-inflammatory immune response mediated by regulatory T cells [166]. Taken together, preclinical trials indicate that hypofractionated dose prescriptions of about 6–12 Gy might shift the balance towards an antitumor immune landscape. Furthermore, Zhang et al., showed similar results using long- and short-term treatment schedules, and they also extended this period into treatment-induced tumor infiltration by T cells with a peak after 5 to 8 days; they concluded that the supply of tumor-specific T cells by regional lymph nodes might be more essential [167].

The TONIC trial investigated immune-induction strategies in metastatic triple-negative breast cancer to enhance sensitivity to PD-1 blockade. Patients receiving 3 × 8 Gy of a single metastatic lesion followed by Nivolumab showed no superior OS compared to nivolumab monotherapy. The highest overall response rate was found after doxorubicin induction. It should be considered that only 12 patients were included in the RT arm, and they showed relatively low PD-L1 expression; moreover, further nivolumab was administered only 2 weeks after RT [168].

As well as housing tumor-infiltrating immune cells, the CNS microenvironment contains immune cells such as tumor-associated macrophages and microglia (TAMs), which are proposed to modulate metastatic colonization and tumor growth. Targeting TAMs with an inhibitor of colony-stimulating factor 1 receptor, as well as the simultaneous use of compensatory STAT5 signaling, showed sustained tumor control and normalization of microglia activation in a pre-clinic murine BCBM model [169]. Another important effector cell type might be tumor-associated myeloid cells (TAMCs), which are highly expressed in PD-L1; these were found to be key drivers of immunosuppression and therapy resistance in glioblastoma. Combining the targeting of TAMCs and RT showed significantly extended survival in a mouse glioma model, highlighting the central role of the CNS immune composition and the effect of RT therein [170].

Considering immunogenic effects, there might be a benefit from a neoadjuvant radiotherapy compared to post-operative radiotherapy of the tumor bed. Patel and colleagues reported a similar local control and distant brain control rate, as well as overall survival, comparing the two approaches in a retrospective analysis. They found a lower rate of radionecrosis, which might be due to less healthy brain tissue irradiated with a better target delineation and the subsequent resection of irradiated tissue. Furthermore, leptomeningeal disease was decreased in the neoadjuvant RT group, with a possible sterilization effect [171]. There are currently two ongoing trials comparing pre- or post-operative SRS for BM (NCT03741673, NCT03368625). In this setting, neoadjuvant SRT combined with induction-immunotherapy could be a promising approach for patients eligible for surgery. The TRIO trial (NCT03978663) is currently investigating three-fraction neoadjuvant irradiation (3 × 8 Gy) of the primary tumor to induce an immuno-oncologic response in patients with local advanced breast cancer, followed by standard treatment.

Taken together, the rapidly growing preclinical and clinical findings on the advantages of combining immunotherapy and radiotherapy do not seem limited to extracranial disease. More research on the right timing and RT regime is essential to obtain the full potential of this approach, and BCBM-specific data are needed due to the immune-privileged CNS side.

## Figures and Tables

**Figure 1 cancers-15-00211-f001:**
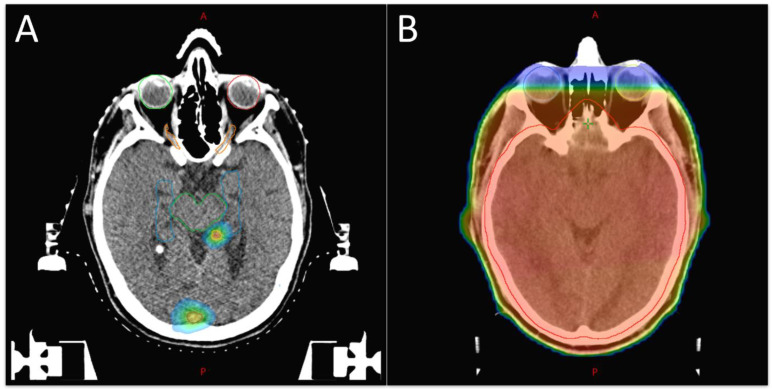
Examples of treatment plans for (**A**) SRT and (**B**) WBRT. Shown are isodose distribution in color wash display (dose fall of from red to blue). (**A**) SRT of two brain lesions with a step dose fall off. Organ at risk delineation with brain stem (green), hippocampus (blue), optic nerves (orange) and eye cavity (light green and red). (**B**) WBRT without hippocampal avoidance. Target volume brain (red).

**Figure 2 cancers-15-00211-f002:**
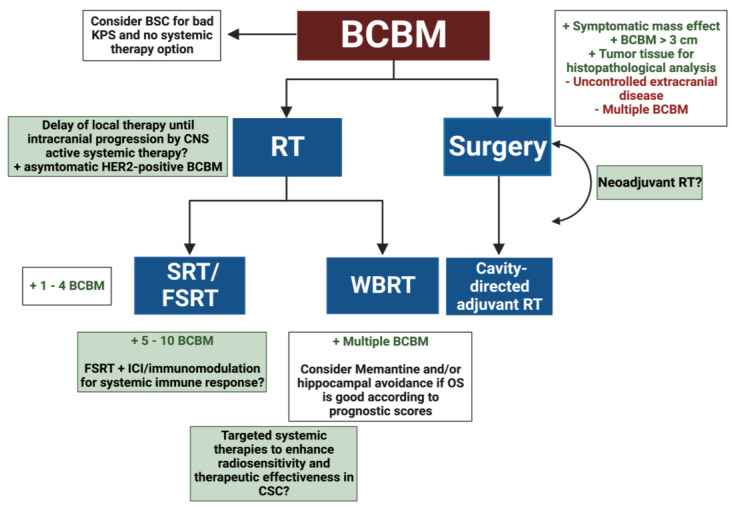
BCBM treatment flowchart adapted from the current ASCO-SNO-ASTRO guideline [68]. The flowchart provides an overview of current therapies (blue boxes) and future options (green boxes) for the treatment of brain metastases in breast cancer with favorable (green letters) and unfavorable outcomes (red letters). Abbreviations: Breast Cancer Brain Metastases (BCBM); Best Supportive Care (BSC); Karnofsky Performance Status (KPS); Whole Brain Radiation Therapy (WBRT); (Fractionated) Stereotactic Radiation Therapy (FSRT/SRT), Radiation Therapy (RT), Immune Checkpoint Inhibitor (ICI), Created with BioRender.com.

**Figure 3 cancers-15-00211-f003:**
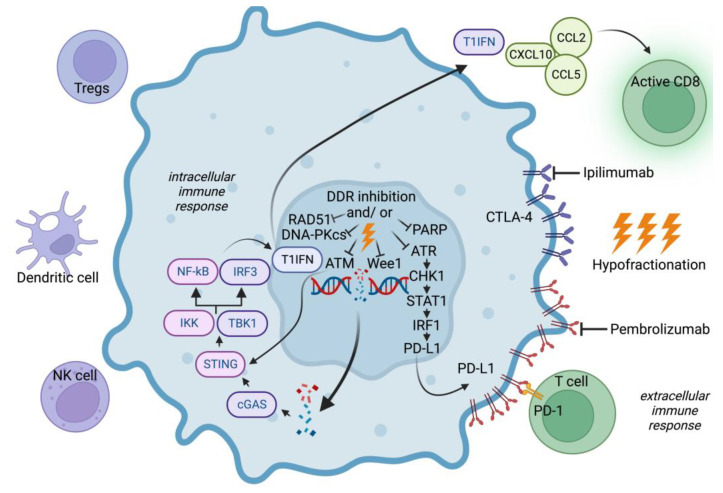
Interplay of DNA damage response inhibition and immune checkpoint inhibition with irradiation to intensify tumor therapy. Inhibition of DDR can lead to DNA damage accumulation, with both double-stranded DNA and single-stranded DNA accumulating in the cytoplasm. Cytoplasmic DNA activates the cGAS/STING pathway and thereby the type I interferon (IFN) pathway, which ultimately activates both chemokines and immune cells (such as T cells, NK cells, and DCs). Specifically, STING promotes phosphorylation and nuclear shift of type I IFN transcriptional regulatory factors TBK1 and IFN regulator 3 (IRF3), while activating the NF-κB pathway that interacts with IRF3. IKK stimulates the type I IFN pathway through downstream transcription factors. Activated TIL releases IFNγ, which acts on tumor cells and mediates STAT1/3-dependent PD-L1 upregulation. The ATM/ATR/Chk1 pathway can also trigger PD-L1 expression. ATM can directly activate and participate in STING-mediated pathways, and PARPi can promote PD-L1 expression by downregulating GSK3β. The NF-κB pathway can be activated to promote transcription and secretion of various pro-inflammatory factors. Following this sequence of events, these factors serve to promote DC activation and elicit an immune response. Radiation sensitization can be achieved by hypofractionated irradiation in combination with the immune checkpoint inhibitors ipilimumab or pembrolizumab. (ATM: ataxia telangiectasia mutated protein; ATR: ataxia telangiectasia and Rad3-related protein; CCL2 or 5: C-C motif chemokine ligand 2/5; cGAMP: cyclic GMP-AMP; cGAS: cyclic GMP-AMP synthase; CHK1: Checkpoint kinase 1; CXCL10: C-X-C motif chemokine ligand 10; CTLA-4: cytotoxic T-lymphocyte-associated protein 4; DDR: DNA damage response; DNA-PKcs: DC: dendritic cell; DNA protein kinase catalytic subunit; IKK: IκB kinase; IRF1: Interferon regulatory factor 1; IRF3: interferon regulatory factor 3; NF-κB: nuclear factor kappa-B; natural killer cell; PARP: poly-ADP-ribose polymerase; PD-1: programmed cell death protein 1; PD-L1: programmed death ligand 1; RAD51: recombinase RAD51; STAT1/3: Signal Transducer and Activator of Transcription 1/3; STING: Stimulator of Interferon Gene; T1IFN: Type I Interferon; TBK1: TANK binding kinase-1; Regulatory T cells; Wee1: Wee1 G2 checkpoint kinase). Created with BioRender.com.

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
