# Peer review of "Radiotherapeutic Strategies to Overcome Resistance of Breast Cancer Brain Metastases by Considering Immunogenic Aspects of Cancer Stem Cells"

_cancers, 2022, doi:10.3390/cancers15010211_

Round 1

Reviewer 1 Report

Hintelmann et al. have summarized and presented an exhaustive review of radiotherapeutic strategies to tackle resistance mechanisms in brain metastases of breast cancer from a clinical perspective. The manuscript has been well-written and the studies mentioned are pertinent to the basic theme of paper. I would recommend this to be accepted after the following minor revisions:

1) It would be helpful to medicinal chemists if structures of small molecules are included as a Figure.

2) There are multiple spelling mistakes throughout the article and the language used is extremely simple and unpolished. Some spelling mistakes include:

Line 790: Please correct the spelling of "microenvironmet" to "microenvironment"; "containes" to "contains"

Line 801: Please correct the spellings of "Consigering" to "Considering"; "immunogetic"to "immunogenic".

Line 802: Please correct the spelling of "radiotherpay" to "radiotherapy".

Line 803: Please correct the spelling of "smilier" to "similar".

Line 804: Please correct the spelling of "approchares" to "approaches".

Line 813: Please correct the spelling of "respondse" to "response"; "locall" to "local"; "adavanced" to "advanced".

3) Please rewrite the paragraph starting from Line 815. Language is very unclear and trivial.

Author Response

Hintelmann et al. have summarized and presented an exhaustive review of radiotherapeutic strategies to tackle resistance mechanisms in brain metastases of breast cancer from a clinical perspective. The manuscript has been well-written and the studies mentioned are pertinent to the basic theme of paper. I would recommend this to be accepted after the following minor revisions:

We thank the reviewer for the favorable review of our review article and have addressed the useful and helpful comments as follows:

1) It would be helpful to medicinal chemists if structures of small molecules are included as a Figure.

We have included Figure 3 in the revised manuscript, which attempts to summarize the complex cellular processes clearly in a simple way.

2) There are multiple spelling mistakes throughout the article and the language used is extremely simple and unpolished. Some spelling mistakes include:

We apologize for the numerous spelling errors and have changed them in the revised version by using the professional editing service from MDPI.

3) Please rewrite the paragraph starting from Line 815. Language is very unclear and trivial.

As requested by the reviewer we have rewritten the paragraph starting from Line 815 as followed:

In pancreatic cancer, Zhang et al. (2019) postulated that activation of the innate immune response through enhanced induction of DNA damage could further increase the efficacy of ICI. They showed that inhibition of ATM alone was able to increase tumoral type I IFN expression independently of the cyclic GMP-AMP synthase (cGAS) and the stimulator of the interferon response CGAMP interactor 1 (STING), but this was only achievable in a manner dependent on tank-binding kinase 1 (TBK1) and the pro-to-oncogenic tyrosine protein kinase Src. The combination of ATM inhibition and IR in-creased TBK1 activity even more markedly and, correspondingly, also increased IFN production and antigen presentation. In addition, the silencing of ATM increased PD-L1 expression and increased the sensitivity of pancreatic tumors to PD-L1-blocking anti-bodies. This was associated with an increase in tumoral CD8+ T cells and established immune memory. The authors further observed that low ATM expression inversely correlated with PD-L1 expression in pancreatic tumors from patients. Overall, these re-sults demonstrate that the efficacy of ICI in pancreatic cancer is enhanced by ATM inhibition and further potentiated by irradiation, depending on the increased immunogenicity of the tumor [147].

Reviewer 2 Report

In this review, authors summarize brain metastases of breast cancer, radioresistance, breast cancer stem cell, and radiotherapeutic strategies for the treatments of the brain metastases of breast cancer. Generally, this review is properly organized and systemically summarizes the literature and clinical trials regarding the treatments combining radiotherapy and targeted therapy and their molecular mechanisms for brain metastases of breast cancer. This summary helps further understanding of brain metastases of breast cancer and highlights the potential of combined therapeutic strategies against brain metastases of breast cancer. Some suggestions for the improvement of this manuscript are provided underneath.

1. In title, the linkage between “immunogenic aspect” and BCBM treatments is not clearly exhibited. Authors may revise the title to better fit their summaries.

2. Section 3 was entitled “Mechanisms of radioresistance in CSCs”; however, “3.3 Immune Evasion of CSC in BCBM” appears to corelate weakly with radioresistance.

3. Several abbreviations are directly used without definition, including CNS and IR in Abstract; BCBM, PBC, PFS and OS in the Introduction, and so on. Many abbreviations are used in this manuscript that may make it difficult to read; however, some of them (e.g. HBEGF, ST6GALNAC5, PDPK1, CCL7, and HDR) appear only once. Authors should consider reducing unnecessary abbreviations.

Author Response

In this review, authors summarize brain metastases of breast cancer, radioresistance, breast cancer stem cell, and radiotherapeutic strategies for the treatments of the brain metastases of breast cancer. Generally, this review is properly organized and systemically summarizes the literature and clinical trials regarding the treatments combining radiotherapy and targeted therapy and their molecular mechanisms for brain metastases of breast cancer. This summary helps further understanding of brain metastases of breast cancer and highlights the potential of combined therapeutic strategies against brain metastases of breast cancer. Some suggestions for the improvement of this manuscript are provided underneath.

  1. In title, the linkage between “immunogenic aspect” and BCBM treatments is not clearly exhibited. Authors may revise the title to better fit their summaries.

According to the good suggestion of the reviewer, we have modified the title of the review article: “Radiotherapeutic Strategies to Overcome Resistance of Breast Cancer Brain Metastases by Considering Immunogenic Aspects of Cancer Stem Cells”

  1. Section 3 was entitled “Mechanisms of radioresistance in CSCs”; however, “3.3 Immune Evasion of CSC in BCBM” appears to corelate weakly with radioresistance.

We agree with the reviewer and have now changed the title for section 3 to: "Mechanisms of Radioresistance and Immune Evasion in CSCs."

  1. Several abbreviations are directly used without definition, including CNS and IR in Abstract; BCBM, PBC, PFS and OS in the Introduction, and so on. Many abbreviations are used in this manuscript that may make it difficult to read; however, some of them (e.g., HBEGF, ST6GALNAC5, PDPK1, CCL7, and HDR) appear only once. Authors should consider reducing unnecessary abbreviations.

We thank the reviewer for the helpful suggestions and would like to apologize for the numerous typographical errors and nonsystematic explanation of abbreviations that made the article unnecessarily difficult to read.

We have now revised the manuscript point by point and corrected all abbreviations and explained them when necessary.

Round 2

Reviewer 2 Report

The previous concerns have been properly clarified and addressed.